# QUANTIZED APPROXIMATELY ORTHOGONAL RECURRENT NEURAL NETWORKS

## ABSTRACT

In recent years, Orthogonal Recurrent Neural Networks (ORNNs) have gained popularity due to their ability to manage tasks involving long-term dependencies, such as the copy-task, and their linear complexity. However, existing ORNNs utilize full precision weights and activations, which prevents their deployment on compact devices.

In this paper, we explore the quantization of the weight matrices in ORNNs, leading to Quantized approximately Orthogonal RNNs (QORNNs). The construction of such networks remained an open problem, acknowledged for its inherent instability. We propose and investigate two strategies to learn QORNN by combining quantization-aware training (QAT) and orthogonal projections. We also study post-training quantization of the activations for pure integer computation of the recurrent loop. The most efficient models achieve results similar to state-of-the-art full-precision ORNN, LSTM and FastRNN on a variety of standard benchmarks, even with 4-bits quantization.

## 1 INTRODUCTION

**Motivation:** Machine learning applications frequently encompass the analysis of time series data, such as textual information and audio signals. Within the realm of deep learning, various Recurrent Neural Network (RNN) architectures and transformers have demonstrated notable success in addressing a diverse array of tasks associated with time series data.

These models typically require a substantial number of parameters for optimal performance and involve numerous matrix-vector multiplications during inference, using matrices and vectors of considerable sizes containing floating-point numbers. This does not allow for the deployment of these networks on compact devices with memory and power constraints, as well as for real-time applications. Overcoming these constraints allows the use of RNNs across a large range of domains in edge-ML and tinyML applications, as described in Abadade et al. (2023), such as healthcare, smart farming, environment, or anomaly detection.

An effective and often unavoidable step[1] to address these challenges is neural network quantization. This technique aims to reduce the number of bits required to represent the weights and activations of the network. As evaluated in Hubara et al. (2018); Gholami et al. (2022), with appropriate hardware and implementation, this accelerates runtime computations, lowers power consumption, and, on the other hand, decreases the amount of space needed for parameter storage. Other authors have gone further and implemented quantized LSTMs on low-cost FPGAs to meet low-power requirements and provide real-time low-cost solutions, see Chen et al. (2024); Bartels et al. (2023).[2]

Our goal is to contribute to the field of quantization of RNNs, with a particular emphasis on considering RNNs able to address tasks involving long-term dependencies such as the copy-task with many time steps. In doing so, we broaden the scope of applications for quantized RNNs.

To achieve this objective, we introduce and compare two strategies for constructing Orthogonal or approximately Orthogonal Neural Networks (ORNNs) with quantized weights. We call them Quantized Orthogonal Recurrent Neural Networks (QORNNs). The orthogonality constraint of

---

[1]For instance, the weights in an Google EdgeTPU are typically encoded as fixed-point numbers using 8 bits.
[2]In these two references, the only compression of the weights is due to quantization.

ORNNs, which has been studied in the articles described in the Appendices A.1 and A.2, have indeed the following advantages:

- Their memory complexity does not increase with the input length, and their computational complexity scales linearly with the input length. They are smaller compared to its competitors, which is ideal for training and inference on long inputs.
- They are easy to learn and have excellent memorization ability, which permits to solve efficiently important tasks with long-term dependencies such as the copy-task.

On the contrary, LSTM and GRU are known to struggle to solve the copy-task with many time steps: performances comparable to a naive baseline, consisting of random guessing, have been reported in Arjovsky et al. (2016); Bai et al. (2018); Tallec and Ollivier (2018); Kerg et al. (2019); Bai et al. (2019). For the copy-tasks studied in Jelassi et al. (2024), the limitation comes rapidly as the length of the sequence increases.

**Contribution:** This paper presents pioneering work in the exploration of the quantization of (ORNN). Our main contributions can be summarized as follows:

- We investigate the factors influencing the impact of quantization on orthogonality and the behavior of ORNNs.
- We propose two different Quantization-Aware Training (QAT) strategies for constructing ORNNs with quantized weights, called Quantized and approximately Orthogonal Recurrent Neural Networks (QORNN).
- We demonstrate that QORNNs are the first quantized recurrent solution that can effectively capture long-term dependencies in the copy-task with $T > 1000$ using only 5 bits for the weights.
- We achieve state-of-the-art results on the permuted pixel-by-pixel MNIST (pMNIST) task, even with 4-bit quantization.
- We further expand our investigation by applying a simple Post-Training Quantization method to the activations, reducing them to 12 bits without any loss in performance. Consequently, we introduce the first fully quantized recurrence capable of solving the copy-task with sequences longer than 1000 steps.

**Organization of the paper:** We discuss articles related to quantized RNNs in Section 2. Descriptions of the main neural networks architecture handling time-series are given in Appendix A.1, and a focus on the articles devoted to ORNNs is in Appendix A.2. The notations and technical descriptions related to vanilla RNNs, orthogonality, and quantization are presented in Sections 3.1, 3.2, and 3.3, respectively. The reasons why quantizing ORNN is unstable is described in Section 3.4. This section also contains bounds on the orthogonality discrepancy of the quantization of orthogonal matrices. The two algorithms for building QORNNs are detailed in Section 4. Finally, experiments and their results are presented in Section 5. Additional details can be found in the appendices. The code implementing the experiments is available at **ANONYMIZED**.

## 2 RELATED WORKS

In this section, we will solely discuss works that describe quantization methods designed for networks manipulating time-series data. Nevertheless, additional bibliographical information on full-precision models for time-series can be found in Appendix A.1. We also offer a comprehensive overview of contributions related to Unitary and Orthogonal RNN[3] in Appendix A.2.

**On quantized RNNs:** The pioneering article on the quantization of weights in RNNs is Ott et al. (2016). In this article, the authors explore the quantization of vanilla RNNs, LSTMs, and GRUs. Then, the existing articles consider the quantization of both weights and activations for LSTM Hou et al. (2017); Nia et al. (2023), LSTM and vanilla RNN (Hubara et al., 2018) or both LSTM and GRU

---

[3]Given that ORNNs achieve comparable performance to URNN (Mhammedi et al., 2017), in the scope of lower complexity, we limit this study to ORNN.

(Zhou et al., 2017; Ardakani et al., 2019; Xu et al., 2018; Alom et al., 2018; Wang et al., 2018). The proposals differ in various aspects including the quantization scheme and the optimization strategy. The performance on the most commonly used tasks is summarized in Appendix A.3.

The article Kusupati et al. (2018) contains the study of a compressed network named fastRNN whose weights are quantized on 8 bits and activations on 16 bits. The architecture of fastRNN contains a skip-connection, similar to the one of ResNET (He et al., 2016), and (optionally) a gating mechanism leading to a model called fastGRNN.

To the best of our knowledge, no article has reported attempts to quantize architectures based on Ordinary Differential Equations, nor on Structured State Space Models (SSSM), see Appendix A.1.

Similarly, we found no articles studying the quantization of ORNNs. The closest studies evaluate quantized vanilla RNNs; see Ott et al. (2016) and Hubara et al. (2018). Both articles emphasize the difficulty of the problem and only provide results for tasks involving short-term dependencies, such as the next character prediction task on the Penn TreeBank (PTB) and text8 datasets. They explain that this difficulty stems from instability.[4] The problem of vanilla RNN quantization is also evoked in the recent survey Gupta and Agrawal (2022).

Finally, this research contributed to the implementation of quantized RNNs and LSTMs on FPGA, as reported in Chen et al. (2024); Gao et al. (2022); Bartels et al. (2023) and the references therein, leading to a drastic reduction in power consumption, latency, and cost.

**Conclusion on RNNs:** Setting aside fastRNN temporarily, as indicated by the performances reported in Appendix A.3 and Gupta and Agrawal (2022), among the quantized RNNs, architectures follow the following general rule

$$\text{LSTM} \gg \text{quantized LSTM} \qquad \text{and} \qquad \text{LSTM} \gg \text{GRU} \gg \text{quantized GRU}$$

where '$\gg$' means 'has better performances than'. For this reason, we compare the QORNNs obtained by the proposed methods to the results of full-precision LSTM, which serves as an optimistic surrogate for all existing quantized LSTM and GRU architectures. We also compare our results to those of fastRNN and fastGRNN (Kusupati et al., 2018). None of the existing quantized RNNs, LSTMs, or GRUs are able to solve the copy-task with many timesteps.

**On quantized Transformers:** The complexity of transformers renders them irrelevant to the scope of the present study, and therefore, we do not delve into this bibliography. However, the article Shen et al. (2020) was the first to address the quantization of weights and activations in BERT, and as described in the recent survey (Tang et al., 2024), many subsequent articles have followed suit.

## 3 PRELIMINARIES AND NOTATIONS

In this section, we provide the main ideas and notations used on the RNN architecture, orthogonality, and quantization.

### 3.1 VANILLA RNNS

Vanilla RNNs define functions that take a time series as input and produce a vector (in the many-to-one case) or a time series (in the many-to-many case) as outputs.

In order to define them, we consider positive integers $n_i$ and $T$, and an input time series $(x_t)_{t=1}^{T} \in (\mathbb{R}^{n_i})^T$ of length $T$, made of $n_i$-dimensional data points. Denoting the output size $n_o \in \mathbb{N}$, the output is either a vector in $\mathbb{R}^{n_o}$ or a time series in $(\mathbb{R}^{n_o})^T$.

The architecture of the RNN is defined by a hidden layer size $n_h \in \mathbb{N}$, an activation function $\sigma$ and an output activation function $\sigma_o$. The parameters defining the vanilla RNN are $(W, U, V, b_o)$ for a recurrent weight matrix $W \in \mathbb{R}^{n_h \times n_h}$, an input-to-hidden matrix $U \in \mathbb{R}^{n_h \times n_i}$, a hidden-to-output matrix $V \in \mathbb{R}^{n_o \times n_h}$, and bias $b_o \in \mathbb{R}^{n_o}$. The hidden-state is initialized with $h_0 = 0$ and then computed using

$$h_t = \sigma\left(W h_{t-1} + U x_t\right) \in \mathbb{R}^{n_h}, \tag{1}$$

---

[4]We illustrate and evaluate this phenomenon in Sections 3.4.

for $t \in [\![1, T]\!]$.

In the many-to-one case, the output of the vanilla RNN is

$$\sigma_o(V h_T + b_o) \in \mathbb{R}^{n_o}.$$

In the many-to-many case, the output of the vanilla RNN is

$$\left( \sigma_o(V h_t + b_o) \right)_{t \in [\![1, T]\!]} \in (\mathbb{R}^{n_o})^T.$$

In all the experiments, $\sigma$ is either the ReLU or the modReLU (Helfrich et al., 2018) activation functions, $\sigma_o$ is the identity function for regression tasks and the softmax function for classification tasks. The parameters $(W, U, V, b_o)$ are learned and $W$ is constrained to be quantized and approximately orthogonal. The matrix $U$ is also quantized.

## 3.2 ORTHOGONALITY

The matrix $W \in \mathbb{R}^{n_h \times n_h}$ is orthogonal if and only if

$$W'W = WW' = I,$$

where $I$ denotes the identity matrix in $\mathbb{R}^{n_h \times n_h}$ and $W'$ is the transpose of $W$. This necessitates that the columns (respectively, rows) of the matrix possess a Euclidean norm of 1, with the additional condition that any two distinct columns (respectively, rows) exhibit a scalar product of 0. Among the various properties of orthogonal matrices, it is important to note that the singular values of orthogonal matrices are all equal to 1. Denoting $\sigma_{\min}(W)$ and $\sigma_{\max}(W)$ as the smallest and largest singular values of $W$, we have $\sigma_{\min}(W) = \sigma_{\max}(W) = 1$. In other words, multiplication by an orthogonal matrix preserves norms. Orthogonal matrices constitute the Stiefel manifold (Edelman et al., 1998) that we denote $St(n_h)$.

$$St(n_h) = \{ W \in \mathbb{R}^{n_h \times n_h} \mid W'W = WW' = I \}.$$

The motivation behind constraining the recurrent weight matrix to be orthogonal is to mitigate instability and prevent issues such as vanishing or exploding gradients. This phenomenon has been discussed in numerous articles, and we reiterate it for completeness in Appendix B.

In the models described in Section 4, we consider two strategies that rigorously impose $W$ to be orthogonal. Both strategies establish a mapping from $\mathbb{R}^{n_h \times n_h}$ to $St(n_h)$.

- **projUNN:** The first strategy employs the mapping $P_{\text{projUNN}}$ as defined and implemented, referred to as projUNN-D, in Kiani et al. (2022). This mapping computes the image $P_{\text{projUNN}}(W)$ of matrix $W$ as the nearest orthogonal matrix in terms of the Frobenius norm. The implementation relies on a closed-form expression derived in Keller (1975): $P_{\text{projUNN}}(W) = W(W'W)^{-\frac{1}{2}}$. In the sequel, we use $P_{\text{projUNN}}$ to implement a projected gradient descent algorithm solving a minimization problem involving an orthogonality constraint.

- **Björck:** The second strategy was introduced in Björck and Bowie (1971); Anil et al. (2019) and applies a fixed and sufficiently large number of iterations of the following recursion

$$A_{k+1} = \frac{3}{2} A_k - \frac{1}{2} A_k A_k' A_k, \text{initialized at } A_0 = \frac{1}{\sigma_{\max}(W)} W$$

  More details are given in Appendix C. The resulting mapping from $\mathbb{R}^{n_h \times n_h}$ to the Stiefel manifold $St(n_h)$, denoted as $P_{\text{Björck}}$, is surjective. Therefore, minimizing $L(P_{\text{Björck}}(W))$ among unconstrained $W$ is equivalent to minimizing $L(W)$ among orthogonal $W$. Notice that standard backpropagation permits to compute $\frac{\partial L \circ P_{\text{Björck}}}{\partial W}\Big|_W$.

## 3.3 QUANTIZATION

We consider in this paper the most common scheme of quantization: a uniform quantization with a scaling parameter (Rastegari et al., 2016; Gholami et al., 2022). For a quantization of bitwidth $k$, where $k \geq 2$, the possible values are restricted to a set of size $2^k$, defined as follows:

$$\text{for a given } \alpha > 0, \quad \mathcal{Q}_k = \frac{\alpha}{2^{k-1}} \left[\!\left[ -2^{k-1}, 2^{k-1} - 1 \right]\!\right].$$

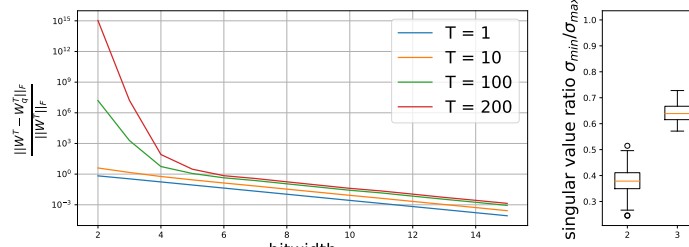

Figure 1: Denote by $q_k$ the quantizer with bitwidth $k$ as defined in Section 3.3, $\sigma_{min}(q_k(W))$ and $\sigma_{max}(q_k(W))$ the smallest and largest singular values of the matrix $q_k(W)$ respectively for $W \in \mathbb{R}^{200 \times 200}$ a uniformly sampled orthogonal matrix. (Left) $\frac{\|W^T - (q_k(W))^T\|_F}{\|W^T\|_F}$ for various $k$ and powers $T$. (Right) Boxplots for 1000 random orthogonal matrices $W$ of the ratio $\sigma_{min}(q_k(W))/\sigma_{max}(q_k(W))$ for various $k$.

The set $\mathcal{Q}_k$ evenly distributes values between $-\alpha$ and $\alpha - \frac{\alpha}{2^{k-1}}$, with a quantization step of $\frac{\alpha}{2^{k-1}}$.

For given $k$ and $\alpha$, the quantizer $q_k$ maps every $W \in \mathbb{R}^{n_h \times n_h}$ to the nearest element in $\mathcal{Q}_k^{n_h \times n_h}$ based on the Frobenius norm. In other words, for every $(i, j) \in [\![1, n_h]\!]^2$, the entry $(i, j)$ of the matrix $q_k(W)$, denoted $(q_k(W))_{i,j}$, is the nearest element in $\mathcal{Q}_k$ to $W_{i,j}$.

When quantizing a matrix $W$, we take the value $\alpha = \|W\|_{\max}$, where $\|W\|_{\max} = \max_{i,j} |W_{ij}|$. For ease of notation, we do not explicitly express the dependence on $\alpha$. Note that, as is common practice (Rastegari et al., 2016), when minimizing a function involving $q_k(W)$ with respect to $W$, we treat $\alpha$ as a constant. Consequently, we do not backpropagate the gradient with respect to $\alpha$. To backpropagate through $q_k$, we employ the classical Straight-Through-Estimator (STE), described for completeness in Appendix D.

## 3.4 QORNN ARE HARD TO TRAIN

**The instability problem:**   We emphasize that quantizing vanilla RNN and ORNN is challenging. It is identified as a difficult unstable problem in Ott et al. (2016); Hou et al. (2017); Hubara et al. (2018). The instability can be attributed to the following phenomena:

- The recurrent weight matrix is applied multiple times, rendering the network's output highly sensitive to even slight variations in the recurrent weight matrix. This also occurs during backpropagation.
- The quantization of an orthogonal recurrent weight matrix generally results in a matrix that is not orthogonal. This, too, can contribute to the instability of the RNN.

Let us illustrate the first point. We present in Figure 1-(Left) the values of $\frac{\|W^T - q_k(W)^T\|_F}{\|W^T\|_F}$ for different bitwidths $k$ and various power $T$ of the matrices[5]. Here, using the method described in Mezzadri (2007), $W \in \mathbb{R}^{200 \times 200}$ is a random matrix sampled according to a uniform distribution over orthogonal matrices of size $200 \times 200$, $\|.\|_F$ represents the Frobenius norm, $q_k$ is the quantization described in Section 3.3, and $T \in \{1, 10, 100, 200\}$. On Figure 1-(Left), we see that $q_k(W)^T$ can be far from $W^T$, especially for small bitwidths $k$ and large timesteps $T$. As a result of this instability, the results of the forward pass using the quantized recurrent weights are far from the results for the full-precision recurrent weights, which may pose challenges in learning the quantized recurrent weights.

We illustrate the second point in Figure 1-(Right): We present boxplots of the ratio $\sigma_{min}(q_k(W))/\sigma_{max}(q_k(W))$, where $\sigma_{min}(q_k(W))$ and $\sigma_{max}(q_k(W))$ are respectively the smallest and largest singular values of $q_k(W)$. We uniformly sample 1000 orthogonal matrices $W$ as described above, apply the quantizer $q_k$ to each of them for different bitwidths $k$, and compute the ratio

---

[5]The analysis made in this paragraph does not take into account the effect of the activation function

| **Algorithm 1** STE-ProjUNN algorithm | **Algorithm 2** STE-Bjorck algorithm |
|---|---|
| **Require:** Initial point: $W_0 \in St(n_h)$, learning rate: $\eta$, projection onto $St(n_h)$: $P_{\text{projUNN}}$ | **Require:** Initial point: $W_0 \in \mathbb{R}^{n_h \times n_h}$, learning rate: $\eta$, map onto $St(n_h)$: $P_{\text{Björck}}$ |
| 1: $i = 0$ | 1: $i = 0$ |
| 2: **while** $x, y \in$ batches **do** | 2: **while** $x, y \in$ batches **do** |
| 3: $\quad q_k(W_i)$ $\qquad\qquad \triangleright$ Quantization | 3: $\quad W_i' = P_{\text{Björck}}(W_i)$ $\qquad \triangleright$ Map onto |
| 4: $\quad L(q_k(W_i), U, V, b_o, x, y) \triangleright$ Forward pass | 4: $\quad q_k(W_i')$ $\qquad\qquad\quad \triangleright$ Quantization |
| 5: $\quad W_i' = W_i - \eta.\nabla L$ $\quad \triangleright$ Weight update | 5: $\quad L(q_k(W_i'), U, V, b_o, x, y) \triangleright$ Forward pass |
| 6: $\quad W_{i+1} = P_{\text{projUNN}}(W_i')$ $\quad \triangleright$ Project | 6: $\quad W_{i+1} = W_i - \eta.\nabla L$ $\quad \triangleright$ Weight update |
| 7: **end while** | 7: **end while** |

$\sigma_{min}(q_k(W))/\sigma_{max}(q_k(W))$. A ratio closer to 1 indicates a higher level of orthogonality. The boxplots show that smaller bitwidths result in smaller expected ratios and more variable ratios.

**Evaluating the approximate orthogonality of $q_k(W)$** Assume $W$ is orthogonal and denote $W_q = q_k(W)$. We obtain the following bounds, proved in Appendix E, for the orthogonality discrepancy of $W_q$:

$$\|W_q W_q' - I\|_F \le 2\frac{n_h}{2^{k-1}} + \left(\frac{n_h}{2^{k-1}}\right)^2. \tag{2}$$

Similarly, we can derive bounds on $\sigma_{\min}(W_q)$ and $\sigma_{\max}(W_q)$:

$$1 - \frac{n_h}{2^{k-1}} \le \sigma_{\min}(W_q) \quad \text{and} \quad \sigma_{\max}(W_q) \le 1 + \frac{n_h}{2^{k-1}}. \tag{3}$$

This permits to obtain guarantees of approximate orthogonality, but only when $n_h \ll 2^{k-1}$, which is not the common setting. Nevertheless, our experiments will show that the following proposed methods are effective in practice, with strategies that enforce both constraints during training, as detailed in the following section.

## 4 QUANTIZED RNNS WITH APPROXIMATE ORTHOGONALITY CONSTRAINTS

In this section, we introduce the two strategies to build QORNN that are evaluated in Section 5. The two strategies can be interpreted as different numerical schemes for solving the same highly non-convex optimization problem, as presented in the following subsections.

### 4.1 PROJECTED STE (STE-PROJUNN)

A QAT strategy is applied to directly learn quantized weights with approximate orthogonality constraints $(q_k(W), q_k(U), V, b_o)$, for a given $k$, where $(W, U, V, b_o)$ are obtained using the *projected gradient descent algorithm* solving the following constrained optimization problem:

$$\begin{cases} \min_{(W,U,V,b_o)} L(q_k(W), q_k(U), V, b_o) \\ W \text{ is orthogonal}, \end{cases} \tag{4}$$

where $L$ is the learning objective. Notice that at each iterate of the algorithm, $W$ is constrained to be orthogonal. In the projected gradient descent algorithm, the gradients are computed using backpropagation and the STE, and the projections onto the Stiefel manifold are computed using $P_{\text{projUNN}}$. Algorithm 1 details each step of STE-projUNN.[6] We use the implementation of the reference code from Kiani et al. (2022) in the $P_{\text{projUNN}}$ repository.

---

[6]Quantization of $U$ matrix is omitted to simplify notations.

## 4.2 STE WITH $P_{\text{BJÖRCK}}$ (STE-BJÖRCK)

A QAT strategy inspired by Anil et al. (2019) is applied to directly learn quantized weights with approximate orthogonality constraints $(q_k(P_{\text{Björck}}(W)), q_k(U), V, b_o)$, for a given $k$, where $(W, U, V, b_o)$ is obtained by a first-order algorithm solving the unconstrained optimization problem:

$$\min_{(W,U,V,b_o)} L(q_k(P_{\text{Björck}}(W)), q_k(U), V, b_o),$$

where $L$ is the learning objective. The gradients are also computed using backpropagation and the STE. Although $W$ is this time unconstrained, as already explained, since $P_{\text{Björck}}$ is surjective onto the Stiefel manifold, solving this problem is equivalent to solving (4). However, the reformulation leads to a different algorithm, a priori facilitating the evolution of the recurrent weight matrix $W$. Besides, the recursive formulation of $P_{\text{Björck}}$ given in Section 3.2 makes it differentiable. Algorithm 2 details each step of STE-Bjorck.

We use the opensource library Deel-Torchlip for $P_{\text{Björck}}$ algorithm implementation.

## 5 EXPERIMENTS

In this section, we present the results of the models described in Section 4 across several standard sequential tasks: the Copy-task in Section 5.1, the permuted and sequential pixel-by-pixel MNIST tasks (pMNIST and sMNIST, respectively) in Section 5.2, and the next character prediction task using the Penn TreeBank dataset in Section 5.3. The first tasks are particularly challenging due to their reliance on long-term dependencies within the sequences, which makes them well-suited for ORNNs. Conversely, the Penn TreeBank task is a language model problem characterized by shorter-term dependencies. Similarly to the next character prediction task, sMNIST is known to be favorable to LSTMs. An additional task, the Adding task, is detailed in Appendix J.

To evaluate the performance of the QORNN, we also compare it with other full-precision RNNs , or when provided by other articles their quantized counterparts:

- LSTM (Hochreiter and Schmidhuber, 1997) which also serves as an optimistic surrogate for all existing quantized models with the exception of FastRNN, see Section 2 and Appendix A.3.
- ORNNs with the same hidden size as the quantized models, implemented using the projUNN-D (Kiani et al., 2022), or Bjorck algorithms.
- FastRNN (Kusupati et al., 2018), either by using the figures from the original article or through additional experiments conducted with the reference code in floating-point.

In the subsequent results, instances where the RNNs did not outperform the naive baseline are denoted by *NC* (Not-Converged).

To verify whether the recurrence of learned QORNN could be fully quantized, we include additional results from a simple Post-Training-Quantization (PTQ) of the activations, detailed in Appendix F.

For each task, model sizes and hyperparameters were selected according to the loss value computed on a validation dataset. A description of the problems, the training hyperparameters, additional results and stability studies are provided in Appendix G, Appendix H, Appendix I and Appendix J.

## 5.1 COPY-TASK

We build RNNs solving the copy-task as described in Wisdom et al. (2016), based on the setup outlined by Hochreiter and Schmidhuber (1997). The input is a sequence of length $T = T_0 + 20$, where the initial 10 elements constitute a sequence for the network to memorize, followed by a marker at $T_0 + 11$. The RNN's objective is to generate a sequence of the same length, with the last ten elements replicating the initial 10 elements of the input sequence. More details are given in Appendix G. This task is known to be a difficult long-term memory benchmark, that classical LSTMs struggle to solve Arjovsky et al. (2016); Bai et al. (2018); Tallec and Ollivier (2018); Kerg et al. (2019); Bai et al. (2019), when $T_0$ is large.

The output activation $\sigma_o$ is the softmax, and the prediction error is measured using the average cross-entropy. The naive baseline has an expected cross-entropy of $\frac{10 \log 8}{T_0 + 20}$.

Table 1: **Performance** of various models and bitwidths for weights and activations on the Copy Task, Sequential MNIST, Permuted MNIST, and Penn TreeBank Character Task (NC stands for 'Not Converged', NU stands for 'Not useful because of other figures', FP for 'full-precision'). Source of figures: [†] from Ardakani et al. (2019); [*] fromKiani et al. (2022) ;[‡] from Kusupati et al. (2018)

| Model | weight bitwidth | activation bitwidth | **Copy-task** cross-ent. | **pMNIST** accuracy | **sMNIST** accuracy | **PTB** BPC |
|---|---|---|---|---|---|---|
| LSTM | FP | FP | NC | 92.00[*] | 98.90[†] | 1.39[†] |
| FastRNN | FP | FP | NC | 90.83 | 96.44[‡] | 1.455 |
| FastGRNN | FP | FP | NC | 92.9 | NU | 1.577 |
| FastGRNN | 8 | 16 | NC | NU | 98.20[‡] | NU |
| STE-Bjorck | FP | FP | 6.4e-6 | 94.51 | 96.61 | 1.404 |
| | 8 | FP | 1.6e-5 | 94.64 | 96.27 | 1.452 |
| | 8 | 12 | 1.7e-5 | 94.76 | 96.20 | 1.452 |
| | 6 | FP | 3.8e-5 | 93.93 | 94.81 | 1.476 |
| | 6 | 12 | 3.7e-5 | 93.94 | 94.74 | 1.476 |
| | 5 | FP | 2.5e-3 | 93.67 | 87.75 | 1.490 |
| | 5 | 12 | 2.5e-3 | 93.67 | 87.70 | 1.490 |
| | 4 | FP | NC | 92.36 | 73.84 | 1.559 |
| | 4 | 12 | NC | 92.33 | 73.38 | 1.559 |
| ProjUNN | FP | FP | 1.1e-12 | 94.3[*] | 90.03 | 1.739 |
| STE-ProjUNN | 8 | FP | 2.0e-10 | 91.27 | 89.53 | 1.742 |
| | 6 | FP | 6.5e-5 | 90.73 | 88.06 | 1.745 |
| | 5 | FP | 1.0e-3 | 90.89 | 87.42 | 1.753 |

As in Kiani et al. (2022), we conducted the experiments for $T_0 = 1000$ timesteps, with ORNNs of size $n_h = 256$. Details on the hyperparameters, learning curves, and results for $n_h = 190$ and $T_0 = 2000$ are provided in the Appendix G.

The fourth column of Table 1 reports the performance for this task. As reported above, LSTM performance remains at the naive baseline level. FastRNN results were not documented in Kusupati et al. (2018). Similarly to what was reported in Kag and Saligrama (2021), none of our experiments with this model achieved convergence, even with full precision weights. Conversely, both STE-projUNN and STE-Bjorck models converge when $k \geq 5$. For $k = 8$ the performance achieved by the QORNN nearly matches that of its floating point counterpart. Our QORNN (configured with $k = 5$ and 12 bits activations) is the first reported RNN with a fully quantized recurrence capable of solving the copy-task for $T_0 = 1000$. Additionally, the size of this QORNN is below 51 kB, see Table 2.

## 5.2 PERMUTED AND SEQUENTIAL PIXEL-BY-PIXEL MNIST (PMNIST/SMNIST)

These tasks are also challenging long-term memory problems. Here, data examples are the $28 \times 28$ images from the MNIST dataset, where each image is flattened to a 784-long sequence of 1-dimensional pixels (normalized values in $[0, 1]$). For pMNIST the pixels are randomly shuffled according to a fixed permutation. The model has to predict the hand-written digit class (10 outputs). Note that pMNIST is a more challenging task for gated models such as LSTM, and is in general not reported in the literature, see Table 4. However it is a classical benchmark task for ORNN.

In this section, we fix $n_h = 170$ for all models. We use the ReLU activation with the STE-Bjorck strategies, and as recommended in Kiani et al. (2022) the modReLU activation with the STE-projUNN, see Appendix H for details.

For sMNIST task, ORNNs in floating point achieve performance comparable to FastRNN, albeit slightly inferior to that of gated models (LSTM and FastGRNN). The STE-projUNN strategy struggles to attain performance levels exceeding 90%. However, the STE-Bjorck model enables to achieve an accuracy of 96.2% with $k = 8$ bitwidth weights quantization and a fully quantized recurrence.

Table 2: **Model sizes** for Copy, MNIST, and Penn TreeBank tasks (Symbol $kP$ stands for 'kilo-parameters'; $kB$ for 'kilo-Bytes'; NC stands for 'Not Converged', NU stands for 'Not useful because of other figures'). Source of figures: [†] from Ardakani et al. (2019); [*] fromKiani et al. (2022) ;[‡] from Kusupati et al. (2018)

| Model | weight bitwidth | **Copy-task** | | **sMNIST** | | **PTB** | |
|---|---|---|---|---|---|---|---|
| | | $kP$ | size ($kB$) | $kP$ | size ($kB$) | $kP$ | size ($kB$) |
| LSTM | FP | NC | NC | 41.4[†] | 162[†] | 4300.8[†] | 16800[†] |
| | 2 | NC | NC | " | 10[†] | " | **525[†]** |
| FastRNN | FP | NC | NC | 30.8 | 120.2 | 1151.0 | 4496 |
| FastGRNN | FP | NC | NC | NU | NU | 1151.0 | 4496 |
| FastGRNN | 8 | NC | NC | 6[‡] | 6[‡] | NU | NU |
| | FP | 70.4 | 275 | 30.8 | 120.2 | 1151.0 | 4496 |
| STE-Bjorck | 8 | " | 75.5 | " | 35.0 | " | 1274 |
| or | 6 | " | 58.9 | " | 27.9 | " | 1005.5 |
| STE-projUNN | 5 | " | 50.6 | " | 24.4 | " | 871.2 |
| | 4 | " | NC | " | 20.8 | " | 737.0 |

For pMNIST and STE-ProjUNN strategy, we could not replicate the projUNN performance reported in Kiani et al. (2022). However, as shown in Table 1, applying STE-Bjorck, with $k = 8$ bitwidth quantization for weight and 12 for activations, provides a QORNN with fully quantized recurrence that achieves results comparable to those reported in Lezcano-Casado and Martínez-Rubio (2019); Kiani et al. (2022), which currently represent the state-of-the-art performance on pMNIST with RNNs. Interestingly, even for $k = 4$, the performance drop remains limited to 2.5% with a network size with less than 21 kByte, see Table 2.

### 5.3 Character level Penn TreeBank

We present the results of QORNNs on a language modeling task. The Penn TreeBank dataset Marcus et al. (1993) (PTB), which comprises sentences of length 150 ($T = 150$), consisting of 50 different characters ($n_i = 50$). The goal of the task is to predict the next character based on the preceding ones (further details can be found in Appendix I). Similar to sMNIST, this task is known to be favorable to LSTMs The purpose of this experiment is to offer a balanced assessment of performance and to evaluate the performance loss on a less favorable task. We also use PTB to perform further ablation studies in Section 5.4 and evaluate the influence of hyperparameters in Appendix I. All experiments described in this section are for $n_h = 1024$, except for the LSTM model where the performance of Ardakani et al. (2019) are reported, 1000 cells. As reported in the literature, we use the Bit Per Character measure (BPC).

The results for this task are reported in the last column of Table 1 and Table 4. The LSTM version with ternary weights proposed in Ardakani et al. (2019), see Table 4, achieves the best results both in performance and size. For the QORNN, the STE-Bjorck strategy's performance is lower by 0.1 BPC with a network size of 872 kBytes.

### 5.4 Ablation study

In this section, we evaluate the influence QAT and projection ($P_{\text{projUNN}}$ or $P_{\text{Björck}}$) in the proposed solutions. We study two additional strategies relaxing one of these constraints (detailed in Appendix K):

- PTQ strategy corresponds to training a full-precision ORNN without any QAT constraint, and applying Post-Training Quantization on the learnt weights for all values of $k$.

- STE-pen strategy imposes soft-orthogonality using a regularization term of the form $\lambda \|q_k(W)(q_k(W))' - I\|_F$, where $\lambda$ is a trade-off parameter. The quantized model is optimized using the STE and does not use any kind of projection.

Table 3: Ablation study: comparison of STE-Bjorck, PTQ, STE-pen on permuted MNIST task

| Task | bitwidth $k$ | STE-Bjorck | PTQ | STE-pen |
|---|---|---|---|---|
| pMNIST | 4 | 92.36 | 13.3 | 44.9 |
| | 6 | 93.93 | 42.2 | 75.4 |
| | 8 | 94.64 | 90.2 | 71.0 |

Table 3 provides a comparison on permuted MNIST and additional results are in Appendix K. PTQ induces a large drop in performance for bitwidths $k < 8$. This illustrates the difficulty of the problem. Moreover, penalization fails to learn effectively across all bitwidths and learning is unstable. This confirms that both QAT and projections are essential for learning QORNN models.

## 6 CONCLUSION AND PERSPECTIVES

In this article, we propose and study two algorithms to construct QORNNs. They enjoy the benefits of ORNNs and work when ORNNs do. In particular, they manage to solve the copy-task for $T_0 = 1000$ and 2000, which existing quantized RNNs were unable to do. We demonstrate that combining orthogonalization and quantization-aware training is crucial for effectively training QORNN. In most experiments, this combination is more efficient when using the Björck orthogonalization method.

Future work on QORNNs could focus: 1/ on implementing QORNNs on dedicated hardware, 2/ on developing learning approaches better taking into account the activation quantization. It is also very much relevant to work on quantizing other models such as SSMs to target longer dependencies such as the Long Range Arena benchmarks described in Tay et al. (2020).

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

## A    BIBLIOGRAPHY COMPLEMENTS

### A.1    NEURAL NETWORKS FOR TIME SERIES

Numerous neural network architectures have been developed specifically for handling time-series data. Rather than attempting to provide an exhaustive overview of all these architectures, our focus is on contextualizing Orthogonal and Unitary Recurrent Neural Networks (ORNN) within this diverse landscape. A comprehensive bibliography on ORNN is deferred to Section A.2.

*Vanilla RNNs* as introduced by Werbos (1988) are notoriously challenging to train due to issues with vanishing and exploding gradients, Werbos (1988); Bengio et al. (1994). This reduces their ability to cope with long-term dependencies. As described in Section 2 and reported in Ott et al. (2016); Hubara et al. (2018); Gupta and Agrawal (2022), quantizing Vanilla RNN is challenging and not efficient.

*LSTM* Hochreiter and Schmidhuber (1997) and *GRU* Cho et al. (2014) are recurrent architectures that tackle vanishing gradient problems by incorporating gating mechanisms. They have demonstrated outstanding performance in tasks such as speech recognition Graves et al. (2013) and neural machine translation Sutskever et al. (2014). LSTM and GRU are known to struggle to solve certain tasks having long-term dependencies, such as the copy-task with many timesteps. On the latter task, comparable performances to a naive baseline, consisting of random guessing, have been reported in Arjovsky et al. (2016); Bai et al. (2018); Tallec and Ollivier (2018); Kerg et al. (2019); Bai et al. (2019). For the copy-tasks studied in Jelassi et al. (2024), the limitation comes rapidly as the length of the sequence increases. To the best of our knowledge, these are the only studies tackling the copy-task using LSTM and GRU architectures. Several variants of quantized LSTM and GRU have been studied in the literature, see Section 2.

*ORNN (and URNN)* are known to achieve superior performance compared to LSTM in handling time series with long-term dependencies Arjovsky et al. (2016). They explicitly address the issues of vanishing and exploding gradients by imposing orthogonality (and unitary) constraints on the recurrent weight matrix of a vanilla RNN, See Appendix A.2. As a bonus, a standard ORNN unit contains about four times fewer parameters than an LSTM unit. To the best of our knowledge, quantization of ORNN has not been previously investigated. A detailed bibliography on full-precision ORNN is in Appendix A.2.

Alternative architectures introduce a *skip connection* in the RNN, similar to the one in ResNet architectures He et al. (2016). Several studies have contributed to the development of this idea Jaeger et al. (2007); Bengio et al. (2013a); Chang et al. (2017). To the best of our knowledge, the only compressed –and therefore quantized– version of this architecture is described in the study by Kusupati et al. (2018).

Due to their ability to capture long-term dependencies and their flexibility, *Transformer architectures* Vaswani et al. (2017) have demonstrated great performance even for long sequences. However, they often require large training datasets and entail computational complexities that render them unsuitable for this study. Several contributions Shen et al. (2020); Prato et al. (2020); Chung et al. (2020); Gupta and Agrawal (2022) have demonstrated that quantization strategies are feasible. A recent survey Tang et al. (2024) is dedicated to this topic.

Many architectures based on an *Ordinary Differential Equation* also benefit from a skip connection and have been studied in several works Chang et al. (2019); Rusch and Mishra (2020; 2021); Lechner and Hasani (2022); Kag et al. (2020); Kag and Saligrama (2021); Erichson et al. (2021). Among these, structured state space sequence models (SSM) Gu et al. (2022) have shown effectiveness in handling tasks with very-long-term dependencies, as described in Tay et al. (2020). To the best of our knowledge, no scientific work has yet studied the quantization of these architectures.

Finally, alternatives that do not fit into the categories described above include the Independent RNN Li et al. (2018), approaches that utilize alternatives to backpropagation Manchev and Spratling (2020); Ororbia et al. (2020), or methods that model infinite-depth networks Bai et al. (2019).

To conclude, ORNNs is a method of choice when the two following properties are simultaneously needed:

- The architecture has a memory complexity independent of the input length and exhibits computational complexity that increases linearly with the input length. This characteristic makes it ideal for training and inference tasks involving long inputs.
- The architecture is easy to learn and has excellent memorization ability, which permits to solve efficiently important tasks with long-term dependencies such as the copy-task with many timesteps.

### A.2 Orthogonal and unitary recurrent neural networks

Learning the unitary and orthogonal recurrent weight matrix of recurrent neural networks has a rich history of study in the last decade. We describe the contributions in chronological order of publication.

Unitary recurrent neural networks were introduced in Arjovsky et al. (2016). In this article, the recurrent weight matrix is the product of parameterized unitary matrices of predefined structures. They argue empirically that URNNs are beneficial because they better capture long-term dependencies than LSTMs. Soon after, the authors of Wisdom et al. (2016) use the *Cayley transform locally* to build an iterative scheme capable of reaching all unitary matrices. In Jing et al. (2017), the authors parameterize the recurrent weight matrix as a product of *Givens rotations* and a diagonal matrix. By doing so, they achieve more efficient models with tunable complexity that can be trained more rapidly. In Mhammedi et al. (2017), the authors parameterize the recurrent weight matrix as a product of *Householder reflections* to reduce the complexity of full-capacity models. In the same line of research, the authors of Jose et al. (2018) explore the use of *product of unitary Kronecker matrices*. They incorporate a soft-orthogonality penalization term to enforce the unitary constraints. The Kronecker architecture can be adjusted to reduce the complexity of the model. In Vorontsov et al. (2017), the authors compare soft and hard-orthogonality constraints. They find that the parameter of the soft-orthogonality strategy under study can be tuned to achieve an approximately orthogonal recurrent matrix, leading to improved efficiency. In Helfrich et al. (2018), the authors narrow their focus to orthogonal recurrent weight matrices and parameterize the entire Stiefel manifold using the *Cayley transform scaled by a diagonal and orthogonal matrix*. Similar to Mhammedi et al. (2017), the number of parameters defining the orthogonal matrix is optimal. The authors of Zhang et al. (2018) use a parameterized Singular Value Decomposition (SVD) to constrain the singular values of the recurrent weight matrix. In Lezcano-Casado and Martínez-Rubio (2019), the authors parameterize orthogonal matrices using the *exponential map*. Finally, in Kiani et al. (2022), the authors develop two Riemannian optimization strategies. The first one is based on the *orthogonal projection onto the Unitary or Stiefel manifold*, and the other on *Riemannian geodesic shooting*. The algorithms are named ProjUNN, and one of them is employed in the presented work. This choice is motivated by the experiments outlined in Kiani et al. (2022).

### A.3 Results in existing articles

In Table 4, we report existing results for full-precision RNNs and RNNs whose weights and activations are quantized for different tasks. The performances obey the general rule

$$\text{fp LSTM} \gg \text{quantized LSTM} \qquad \text{and} \qquad \text{fp LSTM} \gg \text{fp GRU} \gg \text{quantized GRU}$$

where $\gg$ means 'has better performances than' and fp stands for full precision. As a consequence and beside the notable exception of fastRNN, the results for full-precision LSTM provide optimistic surrogates/proxies for the performances of existing quantized models.

## B  Memorization and stability, vanishing and exploding gradient

For a large $T$, if the largest singular value $\sigma_{max}$ of the weight matrix $W$ is smaller than 1, the initial entries of the input $(x_t)_{t=1}^T$ cannot be effectively retained in the hidden state $h_T$. This prevents the consideration of long-term dependencies. Conversely, still considering large $T$, if the smallest singular value $\sigma_{min}$ of $W$ is greater than 1, each multiplication by $W$ in (1) increases the magnitude of $h_t$, and the norm $\|h_t\|$ may tend towards infinity, leading to instability. For memorization and stability issues, it is desirable for the singular values of $W$ to remain close to 1.

We arrive at the same conclusion when attempting to mitigate issues related to vanishing and exploding gradients. As indicated in Arjovsky et al. (2016) and echoed in subsequent literature on URNNs,

Table 4: For every article, the best performance is in bold. The Table illustrates that full-precision LSTM is an optimistic surrogate for all quantized RNNs except fastGRNN and fastRNN. We call skip-RNN the network called FastGRNN-LSQ in Kusupati et al. (2018).

Weights and activation columns: FP stands for 'full precision', t for 'ternary', any number should be interpreted as a bitwidth.

| Task | Reference | T | Model | Weights | Activ. | Score | Metric |
|------|-----------|---|-------|---------|--------|-------|--------|
| PTB word | Zhou et al. (2017) | — | LSTM | FP | FP | **97** | PPW |
| | | | | 2 | 3 | 123 | |
| | | | GRU | FP | FP | 100 | |
| | | | | 4 | 4 | 120 | |
| | Hubara et al. (2018) | 50 | LSTM | FP | FP | **97** | |
| | | | | 4 | 4 | 100 | |
| | Xu et al. (2018) | 30 | LSTM | FP | FP | **89.8** | |
| | | | | 2 | 2 | 95.8 | |
| | | | GRU | FP | FP | 92.5 | |
| | | | | 2 | 2 | 101.2 | |
| | Kusupati et al. (2018) | 300 | RNN | FP | FP | 144.71 | |
| | | | LSTM | FP | FP | 117.4 | |
| | | | skip-RNN | FP | FP | **115.92** | |
| | | | FastGRNN | 8 | 16 | 116.11 | |
| | Wang et al. (2018) | 35 | LSTM | FP | FP | **97.2** | |
| | | | | t | t | 110.3 | |
| | | | GRU | FP | FP | 102.7 | |
| | | | | t | t | 113.5 | |
| PTB char. | Hubara et al. (2018) | 50 | RNN | FP | FP | **1.05** | BPC |
| | | | | 2 | 4 | 1.67 | |
| | Ott et al. (2016) | 50 | RNN | 1 | FP | **1.37** | |
| | Ardakani et al. (2019) | 100 | LSTM | FP | FP | **1.39** | |
| | | | | t | 12 | 1.39 | |
| | | | | 1 | 12 | 1.43 | |
| sequ. MNIST | Ardakani et al. (2019) | 784 | LSTM | FP | FP | **98.9** | Accuracy |
| | | | | t | 12 | 98.8 | |
| | | | | 1 | 12 | 98.6 | |
| | Kusupati et al. (2018) | 784 | LSTM | FP | FP | 97.8 | |
| | | | skip-RNN | FP | FP | **98.72** | |
| | | | FastGRNN | 8 | 16 | 98.20 | |
| | | | FastRNN | 8 | 16 | 96.44 | |

denoting $L$ the loss function, we find that:

$$
\begin{aligned}
\frac{\partial L}{\partial h_t} &= \frac{\partial L}{\partial h_T} \frac{\partial h_T}{\partial h_t} \\
&= \frac{\partial L}{\partial h_T} \prod_{i=t}^{T-1} \frac{\partial h_{i+1}}{\partial h_i} \\
&= \frac{\partial L}{\partial h_T} \prod_{i=t}^{T-1} D_i W',
\end{aligned}
$$

where $D_i = \mathrm{diag}\big(\sigma'(W h_{i-1} + U x_i)\big)$ is the Jacobian[7] matrix of $\sigma$ evaluated at the pre-activation point and $W'$ is the transpose of $W$. If all the singular values of $W$ are less than 1, those of $D_i W'$ are as well, causing the norm of $\frac{\partial L}{\partial h_t}$ to rapidly approach 0 as $t$ decreases—resulting in the vanishing gradient problem. Conversely, if some singular values of $W$ are greater than 1, depending on the activation patterns and $\frac{\partial L}{\partial h_T}$, the norm may explode, leading to the exploding gradient problem. To mitigate these phenomena, it is desirable for the singular values of $W$ to remain close to 1. In other words, we aim for $W$ to be orthogonal or at least approximately orthogonal.

## C  THE BJÖRCK ALGORITHM

Björck algorithm Björck and Bowie (1971) aims to minimize the regularization term

$$
R(W) = \|W W' - I\|_F^2,
$$

As described in Anil et al. (2019), the initialization is done with the matrix $A_0 = \frac{1}{\sigma_{\max}(W)} W$, where $\sigma_{\max}(W)$ is the largest singular value of $W$, computed using the power iteration for a fixed number of iterations. Then it applies a fixed and sufficiently large number of iterations, in practice 15, of the following operation:

$$
A_{k+1} = A_k \left( I + \sum_{i=1}^{p} (-1)^p \binom{-\frac{1}{2}}{p} Q_k^p \right)
$$

where $Q_k = I - A_k' A_k$ and $\binom{z}{p} = \frac{1}{p!} \prod_{i=0}^{p-1}(z-i)$. As described in Anil et al. (2019), we take $p = 1$. In this case, the Björck algorithm corresponds to several iterations of the gradient descent algorithm minimizing $R$:

$$
A_{k+1} = A_k - \frac{1}{2} A_k(A_k' A_k - I) = \frac{3}{2} A_k - \frac{1}{2} A_k A_k' A_k,
$$

initialized at $A_0 = \frac{1}{\sigma_{\max}(W)} W$.

We compute $\left. \frac{\partial P_{\mathrm{Björck}}}{\partial W} \right|_W$ using standard backpropagation but treat $\sigma_{\max}(W)$ as a constant.

## D  THE STRAIGHT-THROUGH-ESTIMATOR

Considering $\alpha$ as fixed, the mapping $W \longmapsto q_k(W)$ is piecewise constant. Its gradient at $W$, denoted as $\left. \frac{\partial q_k}{\partial W} \right|_W$, is either undefined or 0. This issue is well-known in quantization-aware training, which aim to minimize an objective $L(q_k(W))$ with respect to $W$, where $W_q \longmapsto L(W_q)$ is the learning loss. Backpropagating the gradient using the chain rule

$$
\left. \frac{\partial L \circ q_k}{\partial W} \right|_W = \left. \frac{\partial L}{\partial W_q} \right|_{q_k(W)} \left. \frac{\partial q_k}{\partial W} \right|_W
$$

is either not possible or results in a null gradient in this context.

---

[7]Since it is not central to our article, we assume that all the entries of $W h_{i-1} + U x_i$ are non-zero, ensuring that the Jacobian and $\sigma'$ are well-defined.

To address this issue, backpropagation through the quantizer is performed using the straight-through estimator (STE) Hinton (2012); Bengio et al. (2013b); Courbariaux et al. (2015). The STE approximates the gradient using

$$\left.\frac{\partial L \circ q_k}{\partial W}\right|_W \approx \left.\frac{\partial L}{\partial W_q}\right|_{q_k(W)}.$$

When minimizing models that involve $q_k(W)$, we consistently approximate the gradient using the STE and treat $\alpha$ as if it were independent of $W$.

## E  PROOF OF THE BOUNDS (2) AND (3)

Let us first prove (2). Assume $W$ is orthogonal and denote $H = W_q - W$, where $W_q = q_k(W)$. Since $W$ is orthogonal, we have $\|WH'\|_F = \|H'\|_F = \|H\|_F$, and $WW' = I$. Using these equations, we obtain:

$$\begin{aligned}
\|W_q W_q' - I\|_F &= \|(W + H)(W + H)' - WW'\|_F \\
&= \|WH' + HW' + HH'\|_F \\
&\leq 2\|WH'\|_F + \|HH'\|_F \\
&\leq 2\|H\|_F + \|H\|_F^2.
\end{aligned}$$

Considering that, with the quantization scheme defined in Section 3.3, we have $\|H\|_{\max} \leq \frac{\alpha}{2^{k-1}} = \frac{\|W\|_{\max}}{2^{k-1}}$ and noting that, since $W$ is orthogonal, $\|W\|_{\max} \leq 1$, we have the inequalities

$$\|H\|_{\max} \leq \frac{1}{2^{k-1}}. \tag{5}$$

We then deduce that $\|H\|_F \leq n_h \|H\|_{\max} \leq \frac{n_h}{2^{k-1}}$. This leads to

$$\|W_q W_q' - I\|_F \leq 2\frac{n_h}{2^{k-1}} + \left(\frac{n_h}{2^{k-1}}\right)^2$$

and (2) holds.

We prove (3) similarly. We first remark that, using (5), we also have $\sigma_{\max}(H) \leq n_h \|H\|_{\max} \leq \frac{n_h}{2^{k-1}}$ and that, since $W$ is orthogonal, we obtain

$$\sigma_{\min}(W_q) \geq \sigma_{\min}(W) - \sigma_{\max}(H) \geq 1 - \frac{n_h}{2^{k-1}}$$

and

$$\sigma_{\max}(W_q) \leq \sigma_{\max}(W) + \sigma_{\max}(H) \leq 1 + \frac{n_h}{2^{k-1}}.$$

We conclude that (3) holds.

## F  ACTIVATION QUANTIZATION AND COMPLEXITY

**Representation:**  We use classical notations for fixed-point arithmetics. For integers $k \geq 0$ and $l \geq 1$, the set of $l$ bits fixed-point numbers with $k$ bits for the fractional part is denoted

$$Q_{l,k} = \frac{1}{2^k} \left[\!\left[ -2^{l-1}, 2^{l-1} - 1 \right]\!\right] \subset \left[ -2^{l-k-1}, 2^{l-k-1} \right[ \subset \mathbb{R}.$$

When $l = k + 1$, we simply denote

$$Q_k = Q_{k+1,k} = \frac{1}{2^k} \left[\!\left[ -2^k, 2^k - 1 \right]\!\right] \subset [-1, 1[.$$

**Multiplications:**  With these notations, the result of the multiplication of two fixed-point numbers $q \in Q_{l,k}$ and $q' \in Q_{l',k'}$ is such that $q.q' \in Q_{l+l'-1,k+k'}$.

Thus, the result of the multiplication of $q \in Q_k$ by $q' \in Q_{k'}$ is such that $q.q' \in Q_{k+k'+1,k+k'} = Q_{k+k'}$.

**Additions:** We only add fixed-point numbers with the same fractional size $k$. For instance, for $q$ and $q' \in Q_k$ we have $q + q' \in Q_{k+2,k}$.

**Link with weight quantization:** In Section 3.3, we consider a number of bits $k \in \mathbb{N}$ and, for the quantization of the recurrent weights matrix $W \in \mathbb{R}^{n_h \times n_h}$, we consider $\alpha_W = \|W\|_{\max} > 0$ and the set of possible values for the entries of $q_k(W)$ is included in $\alpha_W Q_{k-1}^{n_h \times n_h}$. We write $q_k(W) = \alpha_W \widetilde{W}$, where $\widetilde{W} \in Q_{k-1}^{n_h \times n_h}$.

Similarly, for input weights $U \in \mathbb{R}^{n_h \times n_i}$, we consider $\alpha_U = \|U\|_{\max} > 0$ such that $q_k(U) = \alpha_U \widetilde{U}$, for $\widetilde{U} \in Q_{k-1}^{n_h \times n_i}$.

**Input and Hidden state quantization** The quantized hidden-state $h_t$, for $t \in [\![1, T]\!]$, is encoded using $k_a$ bits, for $k_a \geq 1$. We also consider a fixed $\alpha_h > 0$ such that the quantized hidden-states are $h_t = \alpha_h \tilde{h}_t$, with $\tilde{h}_t \in Q_{k_a-1}^{n_h}$. Given a fixed $\alpha_h > 0$, to avoid the confusion with $q_{k_a}$ whose scaling parameter is variable, we denote by $q_{k_a}^{\alpha_h}(a)$ the closest element of $a \in \mathbb{R}$, in $\alpha_h Q_{k_a-1}$. We extend this definition to vectors.

In practice, $\alpha_h$ needs to be large enough so that $\alpha_h Q_{k_a-1}$ covers the interval of values of the full-precision hidden-state variable entries. We can, however, increase $\alpha_h$ to some extent without sacrificing performance. We will use this possibility later on.

Similarly, we quantize any input $x_t$, for $t \in [\![1, T]\!]$ using $k_i$ bits, for $k_i \geq 1$. For simplicity of notations, we still denote $x_t$ as the quantized inputs. Using a fixed scaling factor $\alpha_i > 0$, we write $x_t = \alpha_i \tilde{x}_t$, where $\tilde{x}_t \in Q_{k_i-1}^{n_i}$.

Again, $\alpha_i$ can be chosen quite freely. In practice, we use the following values.

- For the copy-task and PTB, since the entries of $x_t$ are either 0 or 1, for all $t$, we use $\alpha_i = 2$. Notice that the quantization does not affect the input as soon as $k_i \geq 2$.

- For the two pixel-by-pixel MNIST tasks, since the entries of $x_t$ are normalized 8 bit unsigned integer value in $[0, 1]$, we take $\alpha_i = 1$. The quantization does not affect the input as soon as $k_i \geq 9$.

**Rescaling $q_k(U)$:** It can be shown by induction that, for any real number $\lambda > 0$, and for all inputs $(x_t)_{t=1}^T$, the vanilla RNN of parameters $(W, U, V, b_o)$ using ReLU[8] has the same output as the vanilla RNN of parameters $(W, \lambda U, \frac{1}{\lambda} V, b_o)$.

Indeed, considering an input $(x_t)_{t=1}^T$, denoting $(h_t^\lambda)_{t=1}^T$ the hidden-state variables when using the parameters $(W, \lambda U, \frac{1}{\lambda} V, b_o)$, and using (1), we have $h_1^\lambda = \lambda h_1$, from which we obtain $h_2^\lambda = \sigma(W \lambda h_1 + \lambda U x_2) = \lambda h_2$ etc

In fact, we have for all $t \in [\![1, T]\!]$, $h_t^\lambda = \lambda h_t$. Using $\frac{1}{\lambda} V$ leads to the announced statement.

In the sequel, we use this idea and instead of applying the network of quantized weights $(q_k(W), q_k(U), V, b_o) = (\alpha_W \widetilde{W}, \alpha_U \widetilde{U}, V, b_o)$, for $\widetilde{W} \in Q_{k-1}^{n_h \times n_h}$ and $\widetilde{U} \in Q_{k-1}^{n_h \times n_i}$, we take $\lambda = \frac{1}{\alpha_i \alpha_U}$ and equivalently apply the network of parameters $(\alpha_W \widetilde{W}, \frac{1}{\alpha_i} \widetilde{U}, \alpha_i \alpha_U V, b_o)$.

**The fixed-point arithmetic recurrence:** For simplicity of notation, we drop the exponent $\lambda$ and remind that the quantized hidden-state variable is $h_t = \alpha_h \tilde{h}_t \in \alpha_h Q_{k_a-1}^{n_h}$, for a fixed value of $\alpha_h$ that we will choose later on, and a quantized input $x_t = \alpha_i \tilde{x}_t \in \alpha_i Q_{k_i-1}^{n_i}$. We define the quantized ReLU function by the composition $q_{k_a}^{\alpha_h} \circ \sigma$.

---

[8] We do not provide the details here but this idea can be adapted to modReLU.

Table 5: Value of $\alpha_W$ and $\alpha_h$ for activation quantification across the datasets and bitwidth.

| Model | weight bitwidth | Copy-task | | sMNIST | | pMNIST | | PTB | |
|---|---|---|---|---|---|---|---|---|---|
| | | $\alpha_W$ | $\alpha_W \alpha_h$ | $\alpha_W$ | $\alpha_W \alpha_h$ | $\alpha_W$ | $\alpha_W \alpha_h$ | $\alpha_W$ | $\alpha_W \alpha_h$ |
| STE-Bjorck | 4 | – | – | 0.4338 | 2.0 | 0.3661 | 1.0 | 0.1952 | 1.0 |
| | 5 | 0.2651 | 4.0 | 0.3656 | 2.0 | 0.3444 | 1.0 | 0.2350 | 1.0 |
| | 6 | 0.2818 | 4.0 | 0.4094 | 4.0 | 0.3866 | 1.0 | 0.1661 | 1.0 |
| | 8 | 0.2609 | 4.0 | 0.4073 | 2.0 | 0.6007 | 2.0 | 0.4827 | 1.0 |

The recurrence (1) using parameters $(\alpha_W \widetilde{W}, \frac{1}{\alpha_i} \widetilde{U}, \alpha_i \alpha_U V, b_o)$ and quantized ReLU becomes

$$\alpha_h \tilde{h}_t = h_t = q_{k_a}^{\alpha_h} \circ \sigma \left( \alpha_W \widetilde{W} h_{t-1} + \frac{1}{\alpha_i} \widetilde{U} x_t \right)$$

$$= q_{k_a}^{\alpha_h} \circ \sigma \left( \alpha_W \alpha_h \widetilde{W} \tilde{h}_{t-1} + \widetilde{U} \tilde{x}_t \right)$$

The matrix-vector multiplications $\widetilde{W} \tilde{h}_{t-1}$ and $\widetilde{U} \tilde{x}_t$ can be computed using fixed-point multiplications and additions. We leverage the freedom in choosing $\alpha_h$ to ensure that the multiplication by $\alpha_W \alpha_h$ can be performed with a simple bit-shift. More precisely, to perform the Post-Training Quantization of the activation, given the quantized weights, we first compute $\max_h = \max_{t \in [\![1,T]\!]} \|h_t\|_\infty$, for all the full-precision $h_t$ computed for the inputs in the train and validation datasets. We expect the constraint

$$\alpha_h \geq \max_h \tag{6}$$

to limit the saturation effects of the activation quantization. We finally take for $\alpha_h$ the smallest number satisfying the constraint (6) such that $\alpha_W \alpha_h$ is a power of 2. The values of $\alpha_W \alpha_h$ used in the experiments are given in Table 5.

Finally, all the entries of $\sigma \left( \alpha_W \alpha_h \widetilde{W} \tilde{h}_{t-1} + \widetilde{U} \tilde{x}_t \right)$ belong to finite set whose size depends on $(k, k_a, k_i)$ and $\alpha_W \alpha_h$. We can therefore directly compute $\tilde{h}_t$ using a simple look-up table without any floating point computation.

**Complexity evaluation** Table 6 gives the computational complexities for the matrix-vector multiplications appearing in the recurrent layer of the full-precision *Floating Point* RNN, the RNN whose weights have been quantized, called *Quantized weights*, and the *Fully Quantized* RNN.

For RNNs using quantized weights, i.e. for the third column of Table 6, for instance for $W$, we decompose

$$q_k(W) = \frac{\alpha_W}{2^{k-1}} \left( -2^{k-1} B_{k-1} + \sum_{i=0}^{k-2} 2^i B_i \right)$$

where, for all $i \in [\![0, k-1]\!]$, $B_i \in \{0, +1\}^{n_h \times n_h}$ is a binary matrix. This leads to the complexities in the third column of Table 6.

For the *Fully quantized* network described in this section, we obtain the complexities in the last column of Table 6.

Finally, for the copy-task and PTB, since the inputs $x_t$ are one-hot encoded and therefore binary, the input layer can be computed using only $n_h$ multiplications and $(k_i - 1).(n_i - 1).n_h$ additions in floating point arithmetic, in the third column of Table 6, and fixed-precision arithmetic in last columns of Table 6 respectively.

## G   COMPLEMENTS ON THE COPY-TASK EXPERIMENTS

**Detailed task description:** This task is the same experiment as in Wisdom et al. (2016), based on the setup defined by Hochreiter and Schmidhuber (1997); Arjovsky et al. (2016). The copy-task is known to be a difficult long-term memory benchmark, that classical LSTMs struggle to solve

Table 6: Computational complexity for matrix-vector multiplications in the recurrent layer of the RNN. FP stands for in floating-point arithmetic, fpp stands for fixed-point precision additions, $\text{fpp}_{l,l'}$ stands for fixed-precision multiplications between numbers coded using $l$ and $l'$ bits. We neglect the bit-shifts and the accesses to the look-up table.

| Layer | Operation | Full-precision | Quantized weights | Fully Quantized |
|---|---|---|---|---|
| Input matrix | Mult. | $n_i.n_h$ FP | $n_h$ FP | $n_i.n_h$ $\text{fpp}_{k,k_i}$ |
| | Add. | $n_i.n_h$ FP | $k.n_i.n_h$ FP | $n_i.n_h$ fpp |
| Recurrent matrix | Mult. | $n_h.n_h$ FP | $n_h$ FP | $n_h.n_h$ $\text{fpp}_{k,k_a}$ |
| | Add. | $n_h.n_h$ FP | $k.n_h.n_h$ FP | $n_h.n_h$ fpp |

Arjovsky et al. (2016); Bai et al. (2018); Tallec and Ollivier (2018); Helfrich et al. (2018); Kerg et al. (2019); Bai et al. (2019).

Here, input data examples are in the form of a sequence of length $T = T_0 + 20$, whose first 10 elements represent a sequence for the network to memorize and copy. We use a vocabulary $V = \{a_i\}_{i=1}^{p}$ of $p = 8$ elements, plus a blank symbol $a_0$ and a delimiter symbol $a_{p+1}$. Each symbol $a_i$ is one-hot encoded, resulting in an input time series where $n_i = 10$ and an output time series where $n_o = 9$ ($a_{p+1}$ is not a target value).

An input sequence has its first 10 elements sampled independently and uniformly from $V$, followed by $T_0$ occurrences of the element $a_0$. Then, $a_{p+1}$ is placed at position $T_0 + 11$, followed by another 9 occurrences of $a_0$. The RNN is tasked with producing a sequence of the same length, $T_0 + 20$, where the first $T_0 + 10$ elements are set to $a_0$, and the last ten elements are a copy of the initial 10 elements of the input sequence.

The naive baseline consists of predicting $T_0 + 10$ occurrences of $a_0$ followed by 10 elements randomly selected from $V$. Such a strategy results in an expected cross-entropy of $\frac{10 \log 8}{T_0 + 20}$.

**Hyperparameters:** We use, as in Kiani et al. (2022), 512000 training samples, and 100 test samples.

As described in Kiani et al. (2022), for projUNN-D (i.e. projUNN (FP), and STE-projUNN strategies), we use Henaff initialization Henaff et al. (2016). For all approaches, we use *modReLU* for the activation function $\sigma$ Helfrich et al. (2018).

The initial learning rate is $7e - 4$ for projUNN-D (i.e. projUNN (FP), and STE-projUNN strategies) A divider factor of 32 is applied for recurrent weights update, as described in Kiani et al. (2022). For STE-Bjorck and Bjorck (FP) the initial learning rate is set to $1e - 4$. For all methods, a learning rate schedule is applied by multiplying the learning rate by 0.9 at each epoch. We use the RMSprop optimizer for projUNN-D (i.e. projUNN (FP) and STE-projUNN strategies), applying the projUNN-D algorithm with the LSI sampler and a rank 1 (as described in Kiani et al. (2022)). For STE-Bjorck, we use the classical Adam optimizer. Batch size is set to 128.

The training spanned 10 epochs.

For LSTM (FP), we report the results given by Wisdom et al. (2016), which indicates that in this setting LSTM (FP) remains stuck at the naive baseline.

**Complementary results:** We present on Figure 2 the evolution of test loss during the training for STE-projUNN and STE-Bjorck, for several bitwidths.

We present the results for the copy-task in Table 7 with $T_0 = 1000$ time steps and $n_h = 190$. The conclusions drawn are similar to those depicted in Table 1, where $T_0 = 1000$ and $n_h = 256$. The main difference between the results in Table 7 for $n_h = 190$ and Table 1 for $n_h = 256$ lies in the fact that a smaller value of $n_h$ allows achieving comparable performance but with a larger bitwidth $k$. There appears to be a trade-off between hidden size and bitwidth. Ideally, the trade-off should be optimized in order to diminish the networks size.

We present in Table 8 the results for the copy-task with $T_0 = 2000$ time-steps and $n_h = 256$. This task is more challenging to learn. Both STE-projUNN and STE-Bjorck needs a quantization that uses

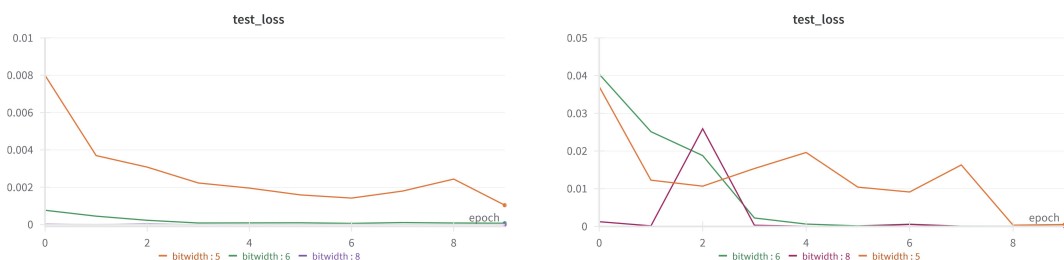

Figure 2: Evolution of Test Loss During Training for the copy-task with $n_h = 256$. (Left) STE-projUNN ; (Right) STE-Bjorck.

Table 7: Performance for STE-Bjorck and STE-ProjUNN for the Copy-task for $T_0 = 1000$ and $n_h = 190$ for several weight bitwidths.

| Model | $T_0$ | $n_h$ | weight bitwidth | | | |
|---|---|---|---|---|---|---|
| | | | FP | 5 | 6 | 8 |
| STE-Bjorck | 1000 | 190 | 8.1e-6 | 1.4e-2 | 1.0e-3 | 3.2e-05 |
| STE-ProjUNN | 1000 | 190 | 6.0e-10 | 2.9e-3 | 6.4e-4 | 2.4e-08 |

6 bits reach a performance below the naive baseline. It seems likely that increasing the hidden size $n_h$ would allow for a reduction in bitwidth.

# H COMPLEMENTS ON THE SMNIST/PMNIST TASK EXPERIMENTS

**Hyperparameters:** We use the $60,000$ training samples and $10,000$ test samples from the MNIST dataset.

As described in Kiani et al. (2022), we employ a random orthogonal matrix initialization for the recurrent weight matrix. The activation function $\sigma$ is *ReLU* for STE-Bjorck, and for PTQ and STE-pen in the ablation study of Section 5.4. We utilize *modReLU* Helfrich et al. (2018) for projUNN-D (i.e. projUNN (FP), STE-projUNN strategies), as performances achieved with ReLU are inferior.

The initial learning rate is $1e - 3$ for all strategies and weights. It remains constant for STE-pen, in the ablation study. A learning rate schedule is applied by multiplying the learning rate by 0.2 every 60 epochs for projUNN-D (i.e. projUNN (FP) and STE-projUNN strategies) and STE-Bjorck.

We utilize the RMSprop optimizer for projUNN-D (i.e. projUNN (FP) and STE-projUNN strategies), implementing the projUNN-D algorithm with the LSI sampler and a rank of 1 (as described in Kiani et al. (2022)). For STE-Bjorck and STE-pen we employ the classical Adam optimizer.

Table 8: Performance for STE-Bjorck and STE-ProjUNN on the Copy-task for $T_0 = 2000$ and $n_h = 256$ for several weight bitwidths.

| Model | $T_0$ | $n_h$ | weight bitwidth | | | |
|---|---|---|---|---|---|---|
| | | | FP | 5 | 6 | 8 |
| STE-Bjorck | 2000 | 256 | 8.9e-6 | NC | 9.4e-3 | 4.3e-05 |
| STE-ProjUNN | 2000 | 256 | 4.2e-11 | NC | 7.1e-4 | 7.5e-06 |

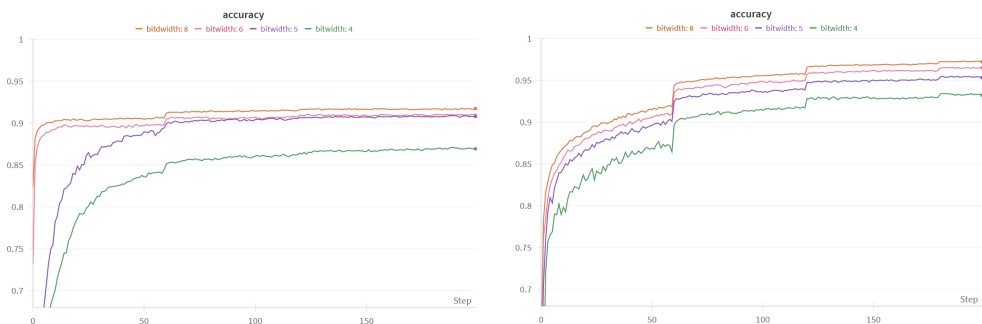

Figure 3: Evolution of the accuracy on the training set during training for pMNIST with $n_h = 170$. (Left) STE-projUNN ; (Right) STE-Bjorck.

Table 9: Performance for STE-Bjorck and STE-ProjUNN on the pMNIST $n_h = 360$ for several weight bitwidths

| Model | $n_h$ | weight bitwidth | | | | |
|---|---|---|---|---|---|---|
| | | FP | 3 | 5 | 6 | 8 |
| STE-Bjorck | 360 | 95.43 | 93.32 | 95.71 | 95.51 | 95.20 |
| STE-ProjUNN | 360 | 93.28 | 44.29 | 92.17 | 93.06 | 93.10 |

For STE-pen, the regularization parameter $\lambda$, which governs the trade-off between optimizing the learning objective and the regularizer (see Equation (7)), is set to $1e - 1$.

The batch size is set to 64 for STE-pen. For all other approaches, it is set to 128.

The training spanned 200 epochs.

LSTM results were taken from Ardakani et al. (2019) for sMNIST and Kiani et al. (2022) for pMNIST.

For FastRNN, results for sMNIST task were given in Kusupati et al. (2018). For pMNIST task, we set the hidden layer size to $n_h = 128$, using *Tanh* and *Sigmoid* as activation function for recurrent and the gate, as described in Kusupati et al. (2018).

Initial learning rate is set to $1e - 3$, and a learning rate schedule is applied by multiplying the learning rate by 0.7 every 60 epochs. Batch size is also set to 128. We also use the classical Adam optimizer.

**Complementary results:** We present on Figure 3 the evolution of training accuracy during the training for STE-projUNN and STE-Bjorck, for several bitwidths.

We present the results for pMNIST in Table 9 with a larger hidden size, $n_h = 360$. The qualitative conclusions drawn are similar to those depicted in Table 1 for $n_h = 170$. Note that STE-Bjorck with $n_h = 360$ achieves an accuracy higher than 93% even for $3 - bits$ quantization but with a model parameters size of 62 kBytes.

## I    COMPLEMENTS ON THE CHARACTER LEVEL PENN TREEBANK EXPERIMENTS

**Detailed task description:** The Penn TreeBank dataset consists of sequences of characters, utilizing an alphabet of 50 different characters. The dataset is divided into 5017K training characters, 393K validation characters, and 442K test characters. Sentences are padded with a blank value when their size is less than the fixed sequence length of 150 characters. The task aims to predict the next character based on the preceding ones. Formally expressed as time series, each sentence represents an input time series of size $n_i = 50$ (since characters are one-hot encoded) with $T = 150$, and the corresponding output time series is identical to the input time series but shifted by one character.

Table 10: Performance for STE-Bjorck and STE-ProjUNN on Penn TreeBank with $n_h = 2048$ and several weight bitwidths

| Model | $n_h$ | weight bitwidth | | | | |
|-------|-------|----|----|----|----|----|
| | | FP | 4 | 5 | 6 | 8 |
| STE-Bjorck | 2048 | 1.45 | 1.53 | 1.45 | 1.43 | 1.45 |
| STE-ProjUNN | 2048 | 1.60 | NC | 1.79 | 1.66 | 1.60 |

The models are evaluated using the Bit Per Character measure (BPC), which is the base-2 logarithm of the likelihood on masked outputs (to exclude padded values for evaluation).

**Hyperparameters:** As described in Kiani et al. (2022), we take a random orthogonal matrix initialization for the recurrent weights. We use *ReLU* as the activation function $\sigma$ for STE-Bjorck. For projUNN-D (i.e., projUNN (FP) and STE-projUNN strategies), we utilize *modReLU* Helfrich et al. (2018) as the activation function, since performances achieved with ReLU were found to be inferior.

The initial learning rate is set to $1e-3$ for all strategies and weight types. A divider factor of 8 is applied to recurrent weight updates for projUNN strategies. A learning rate schedule is implemented by multiplying the learning rate by 0.2 every 20 epochs all strategies.

We employ the RMSprop optimizer for projUNN-D, utilizing the projUNN-D algorithm with the LSI sampler and a rank 1, as described in Kiani et al. (2022). For STE-Bjorck, we use the classical Adam optimizer.

The batch size is set to 128.

The training spanned 60 epochs.

For FastRNN, we set the hidden layer size to $n_h = 1024$, using *Tanh* and *Sigmoid* as activation function for recurrent and the gate, as described in Kusupati et al. (2018).

Initial learning rate is set to $1e-4$, and a learning rate schedule is applied by multiplying the learning rate by 0.7 every 40 epochs. We use the classical Adam optimizer. Batch size is also set to 128. The training spanned 120 epochs.

**Complementary results:** Table 10 presents supplementary results for a larger hidden size of $n_h = 2048$. Most results are similar to those obtained with $n_h = 1024$ for STE-Bjorck. While STE-ProjUNN strategy achieves better results than those with $n_h = 1024$, they are still inferior to the one obtained by STE-Bjorck strategy.

**Influence of hyperparameters:** Table 11 presents the influence of hyperparameters on the performances of the PTB task for a the bitwidth $k = 5$. Experiments were done for other task and other bitwidth with the same conclusions and are not reported.

To obtain the figures on the right of Table 11, we run STE-Bjorck 5 times starting each time from different random initialization. We observe a variation of amplitude 0.01 BPC depending on the random initialization changes. We also observed, but do not report here, that, as expected, the greater the bitwidth, the smaller the variance.

In the middle of Table 11, we see that batch size influences performance, with larger batch sizes generally leading to worse outcomes. This effect may be related to the number of update steps during training and could be mitigated by increasing the number of epochs.

On the left of Table 11, we see that the initial value of the learning-rate impacts the convergence, as it is often the case: very small learning-rates tend to evolve slowly, requiring more epochs, while excessively large rates fail to learn. The range of acceptable learning-rates is reasonably large.

Table 11: Influence of hyperparameters (Learning Rate, Batch size, intialization) on performances of STE-Bjorck for bitwidth $k = 5$

| $k$ | hyperparameters | | | | | | | | | | |
|---|---|---|---|---|---|---|---|---|---|---|---|
| | LR | | | | Batch size | | | | Random initialization | | |
| | 1e-4 | 1e-3 | 1e-2 | 1e-1 | 64 | 128 | 256 | 512 | min | median | max |
| 5 | 1.803 | 1.490 | 1.506 | 2.087 | 1.482 | 1.490 | 1.510 | 1.570 | 1.484 | 1.490 | 1.494 |

## J  COMPLEMENTS ON A REGRESSION TASK: ADDING TASK

**Detailed task description:**  We consider the Adding task as described in Arjovsky et al. (2016). In this task, the input to the RNN is a time series $(x_t)_{t=1}^T \in (\mathbb{R}^2)^T$. Denoting for all $t$, $x_t = (x[0]_t, x[1]_t)$, the sequence $(x[0]_t)_{t=1}^T$ consists of random scalars sampled independently and uniformly from the interval $[0, 1]$, while $(x[1]_t)_{t=1}^T$ consists of zeros except for two randomly selected entries set to 1. The positions of the first and second occurrences of 1 are randomly selected, each following a uniform distribution over the intervals $[\![1, T/2]\!]$ and $[\![T/2 + 1, T]\!]$, respectively. The output is the sum of the two scalars from the first sequence, located at the positions corresponding to the 1s in the second sequence: $\sum_t x[0]_t \cdot x[1]_t$. As $T$ increases, this task evolves into a problem that requires longer-term memory. Naively predicting 1 (the average value of the sum of two independent random variables uniformly distributed in $[0, 1]$) for any input sequence yields an expected mean squared error (MSE) of $\approx 0.167$, serving as our naive baseline.

**Hyperparameters:**  We follow Helfrich et al. (2018) for most settings and consider $T = 750$. We use as in Kiani et al. (2022), 100000 training samples, and 2000 test samples.

As described in Kiani et al. (2022), for all models, the activation function $\sigma$ is the Rectified Linear Unit (ReLU), and $\sigma_o$ is the identity. The recurrent weight matrix is initialized to the identity matrix $I$.

The initial learning rate is $1e - 4$ for projUNN-D (i.e. projUNN (FP) and STE-projUNN strategies), and $1e - 3$ for STE-Bjorck. A divider factor of 32 is applied for recurrent weights update for projUNN, as described in Kiani et al. (2022). A learning rate schedule is applied by multiplying the learning rate by 0.94 at each epoch. We use RMSprop optimizer for projUNN-D (i.e. projUNN (FP) and STE-projUNN strategies) method, applying the projUNN-D algorithm with the LSI sampler and a rank 1 (as described in Kiani et al. (2022)). For STE-Bjorck we use the classical Adam optimizer.

Batch size is set to 50.

The training spanned 50 epochs.

Note that, when learning with projUNN, the recurrent weight matrix remains very close to the identity during the learning process. Since the quantization of such matrices would result in reverting to the identity matrix, we have modified the quantization scheme for this experiment with STE-projUNN strategies . The quantized matrix is defined as $W_q = I + q_k(W - I)$.

All the performances are in Table 12 and Table 13.

In Table 12, we see the results for LSTM (FP) are aligned with those reported in Helfrich et al. (2018), demonstrating its capability to learn this task even over 750 time steps.

In Table 12, we observe that both STE-Bjorck and STE-projUNN achieve a lower MSE than the naive baseline, even with only 2 or 3 bits. However, the STE-projUNN strategy is more challenging to learn than STE-Bjorck. This is possibly due to the resulting matrices being close to the identity.

We present on Table 13 the test accuracy for a larger hidden size $n_h = 400$, STE-projUNN and STE-Bjorck. Conclusions are similar to the ones established for $n_h = 170$. When compared to the results displayed on Table 12, the results of STE-projUNN and STE-Bjorck almost systematically improve. In particular, STE-Bjorck significantly beats the naive baseline even for $k = 2$.

Table 12: Performance for STE-Bjorck and STE-ProjUNN on Adding-task $T = 750$ with $n_h = 170$ and several weight bitwidths (naive baseline is 0.167).

| Model | $n_h$ | weight bitwidth | | | | | |
|---|---|---|---|---|---|---|---|
| | | FP | 2 | 3 | 5 | 6 | 8 |
| LSTM | 170 | 1.0e-4 | | | | | |
| STE-Bjorck | 170 | 9.0e-3 | 0.153 | 0.065 | 0.040 | 0.034 | 8.8e-3 |
| STE-ProjUNN | 170 | 2.0e-4 | 0.170 | 0.165 | 0.080 | 0.147 | 0.062 |

Table 13: Performance for STE-Bjorck and STE-ProjUNN on Adding-task $T = 750$ with $n_h = 400$ and several weight bitwidths (naive baseline is 0.167).

| Model | $n_h$ | weight bitwidth | | | | | |
|---|---|---|---|---|---|---|---|
| | | FP | 2 | 3 | 5 | 6 | 8 |
| STE-Bjorck | 400 | 5.3e-4 | 0.072 | 0.076 | 0.029 | 0.018 | 4.4e-3 |
| STE-ProjUNN | 400 | 2.0e-4 | 0.164 | 0.167 | 0.083 | 0.097 | 0.043 |

## K    COMPLEMENTS ON THE ABLATION STUDY

### K.1    POST-TRAINING QUANTIZATION (PTQ)

For any value of $k$, the weights with approximate orthogonality constraints, and quantized using $k$ bits, are $(q_k(W), q_k(U), V, b_o)$, where $(W, U, V, b_o)$ is the full-precision parameters obtained using the *projUNN-D* algorithm Kiani et al. (2022) for solving

$$\begin{cases} \min_{(W,U,V,b_o)} L(W, U, V, b_o) \\ W \text{ is orthogonal,} \end{cases}$$

where $L$ is the learning objective.

### K.2    PENALIZED STE (STE-PEN)

A Quantized-Aware-Training (QAT) strategy is applied to directly learn quantized weights $(q_k(W), q_k(U), V, b_o)$, with approximate orthogonality constraints, for a given $k$. The weights $(W, U, V, b_o)$ are obtained using an implementation of the Straight-Through Estimator (STE) to solve the following optimization problem:

$$\min_{(W,U,V,b_o)} L(q_k(W), q_k(U), V, b_o) + \lambda R(q_k(W)). \tag{7}$$

Here, $L$ represents the learning objective, $R$ is the regularization term enforcing orthogonality as defined by

$$R(W) = \|WW' - I\|_F^2,$$

and $\lambda$ is a parameter that balances the trade-off between minimizing $L$ and $R$.

For the experiment reported in Section 5.4, the regularization parameter $\lambda$, which governs the trade-off between optimizing the learning objective and the regularizer is set to $1e - 1$.

## L    COMPLEMENTS ON COMPUTATION TIME

Table 14 presents the computation time per epoch for each task and each model. Experiments where done on a NVIDIA GeForce RTX 3080 GPU.

Table 14: Computation time for different models and several tasks

| Task | Model | $n_h$ | $T$ | epoch compute time (minutes) |
|---|---|---|---|---|
| **Copy-task** | STE-Bjorck | 256 | 1020 | 38 |
| | STE-ProjUNN | 256 | 1020 | 39 |
| **pMNIST** | FastRNN | 170 | 784 | 3.8 |
| | FastGRNN | 170 | 784 | 4.2 |
| | STE-Bjorck | 170 | 784 | 2.2 |
| | STE-ProjUNN | 170 | 784 | 4.2 |
| **PTB** | FastRNN | 1024 | 150 | 0.57 |
| | FastGRNN | 1024 | 150 | 0.86 |
| | STE-Bjorck | 1024 | 150 | 1.07 |
| | STE-ProjUNN | 1024 | 150 | 0.37 |

