# OpenReview forum: "Quantized Approximately Orthogonal Recurrent Neural Networks"
_ICLR.cc/2025/Conference — Submitted to ICLR 2025_

### Official Review · Reviewer_cZgn · 2024-11-01

**Soundness:** 3
**Presentation:** 4
**Contribution:** 1
**Rating:** 1
**Confidence:** 5

**Summary:**

This paper introduces Quantized Orthogonal Recurrent Neural Networks (QORNNs) to address the memory and computational limitations of traditional Orthogonal Recurrent Neural Networks (ORNNs) on compact devices. While ORNNs are valued for their capability to handle long-term dependencies in tasks like the copy-task, their reliance on full-precision weights makes them unsuitable for deployment in resource-constrained environments. The authors propose two methods for quantizing ORNNs: Quantization-Aware Training (QAT) with orthogonal projections and post-training quantization for activations, ensuring both efficiency and stability in the recurrent weight matrices. These methods allow QORNNs to maintain the benefits of orthogonality while using fewer bits for weight representation.

**Strengths:**

This a very well-written paper with an extensive literature review, a great introduction to ORRNs and their challenges, many experiments and a thorough appendix with many implementation details for reproducibility

**Weaknesses:**

There is unfortunately not any real innovation in the paper. The only contribution of the paper is applying QAT with a straight-through estimator (a very old idea in quantization literature) to existing optimization methods for learning ORNNs. In fact, the QAT technique is not even state-of-the-art as they could learn the quantization ranges $\alpha$ through gradient by adopting LSQ$^{[1]}$ or PACT^{[2]}.

The theoretical analysis of the impact of quantization in orthogonal matrices is not optimal. It is well known that using MinMax quantization for low-bit quantization (< 6 bits) leads to significant degradation. This is why search-based methods are normally adopted that find the scale or range that minimizes the Forbenious ( or other norms) between quantized and unquantized vectors (commonly known as MSE-based ranges). It would be more interesting to see the plots of Figures 1 & 2 using optimal $\alpha$ values rather than ones based on the maximum range.

[1] LSQ: Learned Step Size Quantization, Esser et al.
[2] PACT: Parameterized Clipping Activation for Quantized Neural Networks

**Questions:**

Why not further reduce the precision of activations by adopting static learned scales akin to LSQ or PACT?

---

### Official Review · Reviewer_LdC4 · 2024-11-01

**Soundness:** 2
**Presentation:** 2
**Contribution:** 2
**Rating:** 5
**Confidence:** 3

**Summary:**

This paper explores the quantization of the weight matrices in Orthogonal Recurrent Neural Nets (ORNNs). Authors introduce and compare two strategies for constructing ORNNs or approximately orthogonal RNNs with quantized weights (QORNNs): STE-projUNN and STE-Björck. These strategies are extensions of two methods of constructing full-precision ORNNs, respectively projUNN and Björck, that employ quantization-aware training with Straight-through assumption (STE).

QORNNs are evaluated on a synthetic copy-task, PTB and few variants of MNIST (pMNIST, sMNIST) datasets and compared against their floating-point variants and other families of RNNs, including LSTM, GRU, fastRNN, fastGRNN. The most efficient models achieve results similar to state-of-the-art full-precision ORNN, LSTM and FastRNN on a variety of standard benchmarks, even with 3-bits quantization.

**Strengths:**

* Seems fairly easy to implement.
* Authors thoroughly motivate and explain challenges of constructing and training ORNNs & instabilities caused by quantization.
* A comparison of model sizes in resulting QORNN models vs. other methods.

**Weaknesses:**

* The proposed methods seem very straightforward: combine the strategy of constructing ORNNs with STE, which is a standard and well-known technique in the quantization literature.
* L286-L300: authors derive bounds for approximate orthogonality of q(W) which they themselves state are too loose to be useful in practice. It would be quite insightful to track and report $||W_q W_q^T – I ||$ during training, to see if the reason of proposed method working well in practice is due to $q(W)$ being fairly close to being orthogonal or not.
* L079: authors claimed SotA results on pMNIST, however they did not compare against other 4-bit sequence-to-sequence models, for instance SSM models such as Mamba [1], transformer models such as LLaMA-3 [2] etc.
* Considered benchmarks are very small by today standards. While most of them seem standard in ORNN literature, it would make a story more convincing if authors included some of the more recent real-world datasets and benchmarks. For instance, it would be insightful to evaluate the proposed methods on some of common reasoning language tasks (MMLU, HellaSwag, Winogrande).

[1] Gu et al., “Mamba: Linear-Time Sequence Modeling with Selective State Spaces”. ArXiV: 2312.00752

[2] Dubey et al., “The Llama 3 Herd of Models”. ArXiV: 2407.21783

**Questions:**

* In Tables 1-2, could you explain what do you mean with “NU” (‘Not useful because of other figures’)?
* What is the memory increase during training for the proposed methods compared to FastRNN, LSTM and vanilla RNN?
* In Table 1, it would be nice if authors could also include accuracy for copy-task, as it is easier to interpret the differences compared to cross-entropy numbers.
* Any reasoning why STE-Björk is faster than FastRNN and FastGRNN on pMNIST in Table 14? It seems quite surprising given that at every optimizer step Björck procedure requires running 15 iterations of the recursion with subsequent backpropagation through this procedure. While it's not the case for some other tasks, providing a detailed analysis of the computational complexity or runtime breakdown for STE-Björk vs. FastRNN/FastGRNN might be quite insightful.

---

### Official Review · Reviewer_R7W2 · 2024-11-02

**Soundness:** 2
**Presentation:** 2
**Contribution:** 2
**Rating:** 3
**Confidence:** 3

**Summary:**

This paper proposes a Quantized Approximately Orthogonal Recurrent Neural Network (QORNN), which is the quantization of Orthogonal Recurrent Neural Network. To address the inherent instability of quantizing vanilla RNN and ORNN, two quantization-aware training strategies are adopted. The method achieves impressive results on a variety of standard benchmarks.

**Strengths:**

The paper is easy to read, and the proposed QORNN is effective even at long-term dependency tasks.

**Weaknesses:**

- There are awkward and missing citations throughout the paper. For example, the citation for fastGRNN should have been first on line 114 instead of line 135. Besides, I think the representation, such as "activations for LSTM Hou et al. (2017)" in line 105, "Penn TreeBank dataset Marcus et al. (1993) (PTB)" is awkward.
- There are some issues on writing (see minor issues and questions below)

Minor Issues
1) There are some minor points on writing:
- Line 90: “The reasons .. is” -> “The reasons .. are”
- Lines 122 and 321: footnotes are written incorrectly
- Footnote 4: “Sections 3.4” should be revised correctly
- Equation in line 203, footnote 5, caption of Table 3, and line 466: please add a comma to the end of the sentence
- Line 286: add a colon to the next of the bolded sentence
- Subtitle 3.4: “QORNN are” -> “QORNN is” or “QORNNs are”
- Tabels 1 and 2, “fromKiani et al. (2022)” -> “from Kiani et al. (2022)”
- Table 2, “sizes for Copy, MNIST” -> “sizes for Copy-task, sMNIST”
2) Are the words "steps", "power", "timesteps" and "length" the same meaning? Mixed terms can confuse the reader. I recommend revising them for clarity. Other examples include copy/copy-task, SSMs/SSM/SSSM, etc.

**Questions:**

1) Please provide a more detailed description of the experimental environment.
2) What is the meaning of "STE-Bjorck or STE-projUNN" in Table 2? I think the experimental results of each model should be separated for clarity. This could be improved by separating the experimental results of each model into different columns.
3) Why is STE-projUNN absent from Table 3?

---

> ### Comment · Reviewer_R7W2 · 2024-12-02
>
> It appears that the authors may have decided not to proceed with addressing the required revisions or feedback. Therefore, I would like to bring the review process for this submission to a close.

---

### Official Review · Reviewer_4gTZ · 2024-11-04

**Soundness:** 2
**Presentation:** 3
**Contribution:** 1
**Rating:** 3
**Confidence:** 4

**Summary:**

In this paper, the authors study the quantization of orthogonal recurrent neural networks (ORNNs). They first investigate the impact of quantization on the orthogonality, including effects on the singular values of the quantized weights and deriving bounds for them. Then the authors propose and investigate two flavors of quantized orthogonal recurrent neural networks (QORNNs) which combine traditional ORNN training with quantization-aware training (QAT). In the experiments they show that QORNNs are competitive with SOTA ORNN, LSTM and FastRNN on a various benchmarks such as copying, flavors of MNIST and PTB.

**Strengths:**

* The investigation on weight quantization effects on ORNNs is interesting and insightful (sec 3.4).
* Based on the experiments the proposed QRNNs (especially with STE-Bjorck) seem to perform well, even till 5-6 bits.
* The paper is clearly written and easy to follow (except a few minor points).

**Weaknesses:**

* The biggest shortcoming of this paper is the limited novelty. Several points regarding this:
    * The authors combine two off the shelf ORNN training algorithms with the most simple (and arguably outdates, see later) flavor of QAT. In other words, they add $q_k(W_i)$ to these algorithms (cf line 3, 4 in algorithms 1, 2, respectively) and assume STE.
    * While I found the investigation in sec 3.4 interesting (see above), I would have expected that these insights come back in their algorithm design (or at least that the authors evaluate such metrics for their proposed approach, look at question whether these metrics in practice correlate with QORNN performance etc).
* On the quantization side, they seem to miss/ignore a lot of innovation that happened in the last 5ish years which could help to potentially have much better performance at low bit-widths. Most importantly:
    * They argue keeping the scale fixed is common practice (line 241/242). Since the introduction of LSQ [1] this is long not common practice anymore.
    * It is unclear to me why the authors not consider per-channel weights. As per-channel weights still work with full fixed-point/integer arithmetic [2], this is supported by almost any HW and commonly used by most quantization literature in the past years. This adds an additional degree of freedom that might be very helpful for ORNNs as by changing the scale ($\alpha$), as one could ensure that rows (or columns, depending on notation) still have a norm of 1 which seems important (cf sec 3.2).
* The proposed QORNNs are actually not orthogonal as the name or text suggest. Only the latent weights ($W_i$) are orthogonal, but the actual quantized weights used for inference ($q_k(W_i)$) are only approximately orthogonal. As the authors themselves show (cf figure 1, sec 3.4), this doesn’t give any guarantees and could be detrimental.
* As the paper positions itself more as ‘explorative’ (and QAT contribution is very limited), I would expect that they also more closely explore PTQ approaches. There are several degrees of freedom that are unexplored, e.g. setting the correct scale (alpha) or applying/adapting common PTQ algorithms such as AdaRound [3] or GPTQ [4].
* Minor:
    * The experimental evaluation is limited to only ‘toy-sized’ datasets/tasks.
    * While it is nice they obtain bounds for q_min/max, the established bounds are so loose that from a practical perspective such bounds are not useful (nor similar to the earlier study they are used to design or evaluate the algorithm).
    * I do miss a comparison to challenges in quantizing transformers or SSMs. While it is arguable whether comparing to transformers it out of the scope of such a paper (as the authors claim), at least discussing/comparing whether the challenges in ORNNs are similar or different to transformers/SSMs would be helpful (e.g. do they suffer from similar outliers as observed in transformers, cf. [5,6]).
    * Regarding SSMs, there is some recent work that would be interesting to compare to [7, 8].

**References:**
* [1] Esser et al., ICLR 2020, Learned Step Size Quantization.
* [2] Nagel et al., 2021, A White Paper on Neural Network Quantization.
* [3] Nagel et al. ICML 2020, Up or Down? Adaptive Rounding for Post-Training Quantization.
* [4] Frantar et al., ICLR 2023, Gptq: Accurate post-training quantization for generative pre-trained transformers.
* [5] Bondarenko et al., EMNLP 2021, Understanding and Overcoming the Challenges of Efficient Transformer Quantization.
* [6] Dettmers et al., NeurIPS 2022, LLM.int8(): 8-bit Matrix Multiplication for Transformers at Scale.
* [7] Pierre et al., ES-FoMo 2024, Mamba-PTQ: Outlier Channels in Recurrent Large Language Models
* [8] Abreu et al, NGSM 2024, Q-S5: Towards Quantized State Space Models

**Questions:**

* It is unclear from the paper why you quantize U but not V. Could you please elaborate on that choice?
* It is unclear to me why the ProjUNN is much better in the copy task but worth on MNIST and PTB (both quantized and unquantized model). Do you have any idea or intuition why this is the case?


**Comments on presentation:**
* $\sigma$ is a bit overuse ($\sigma, \sigma_o$ but also $\sigma_{min}, \sigma_{max}$ for eigenvalues, …)
* The authors reuse $T$ in line 258 again as a power of the weight matrix, which seem based on the text has not any connection to the sequence length T. I suggest the authors either use a different symbol or explain why to the power of $T$ would be relevant here.
    * I can see that this likely comes from the sequence length $T$, though given there is activation function $\sigma$ (and $Ux$) there, it is unclear why this would be important or relate to the final task loss. If this is the case, please elaborate.
* Many equations do not have numbers which makes it hard for readers to refer to them (even if authors do not refer themselves to some equations, a future reader might want to do so). Therefore I suggest to follow common best practices and number all equations that are not in-line.

---

### Meta-Review · Area_Chair_fbeh · 2024-12-25

**Metareview:**

There were serious concerns raised by the reviewers related to the lack of novelty, poor writing, and experimental evaluations.

The authors did not reply to those concerns during the rebuttal period. Since the paper did not receive promising reviews to begin with and the authors did not reply to the concerns raised, I am proceeding with the judgement of the reviewers and suggesting rejection of this work in its current form.

**Additional Comments On Reviewer Discussion:**

The reviewers did a very good job in providing thorough reviews to this work. I hope these reviews will help the authors in improving their work for any potential future submissions.

The authors did not engage during the rebuttal period.

---

### Decision · Program_Chairs · 2025-01-22

Reject